# Modeling the Impact of Mentoring on Women's Work-Life Balance: A Grounded Theory Approach

Parvaneh Bahrami [1], Saeed Nosratabadi [2], Khodayar Palouzian [3] and Szilárd Hegedűs [4,*]

1 Department of Management, Faculty of Management and Accounting, Allameh Tabatabai University, Tehran 1489684511, Iran
2 Doctoral School of Economic and Regional Sciences, Hungarian University of Agriculture and Life Sciences, 2100 Gödöllő, Hungary
3 Department of Management, University of Tehran, Tehran 1417935840, Iran
4 Department of Finance, Faculty of Finance and Accountancy, Budapest Business School, 1149 Budapest, Hungary
* Correspondence: hegedus.szilard@uni-bge.hu

**Abstract:** The purpose of this study was to model the impact of mentoring on women's work-life balance. Indeed, this study considered mentoring as a solution to create a work-life balance of women. For this purpose, semi-structured interviews with both mentors and mentees of Tehran Municipality were conducted and the collected data were analyzed using constructivist grounded theory. Findings provided a model of how mentoring affects women's work-life balance. According to this model, role management is the key criterion for work-life balancing among women. In this model, antecedents of role management and the contextual factors affecting role management, the constraints of mentoring in the organization, as well as the consequences of effective mentoring in the organization are described. The findings of this research contribute to the mentoring literature as well as to the role management literature and provide recommendations for organizations and for future research.

**Keywords:** mentoring; women studies; work-life balance; role management; grounded theory





## 1. Introduction

One of the most important requirements for people to have a good job performance is their ability to meet the multiple and sometimes conflicting expectations that are expected of a person who is in different roles at the same time. The existence of ambiguity and incompatibility in roles has adverse consequences on the performance and physical and mental health of employees. One of the main roles that people play is their simultaneous role in the organization and family (Kooli and Muftah 2020). The need to play an effective role in these two areas has been raised in the concept of work-ife balance. The concept of work-life balance is, initially, defined in terms of the conflict between work and personal life. The concept of work-life conflict occurs when the collective demands of two different roles of individuals in the organization are incompatible. Thus, employee engagement in a role creates problems for other roles they play (Kumara and Fasana 2018). Work-life balance is the perception of employees of work and non-work activities that are in line with current life priorities and lead to the growth and development of individuals (Gragnano et al. 2020). The concept of work-life balance of employees has received a lot of attention. The most important reasons for this have been the change in society's views on gender roles and the consequent increase in the number of women in the workplace (Najam et al. 2020). Women make up almost half of the population of any society and are the nurturers and supporters of future generations. Their presence and participation in the workplace are one of the most important social changes in the last century. As the business world grows with competition, access to a diverse and quality workforce is critical for organizations.

Women can play a significant role in the management of organizations, and this diversity in human resources in an organization can ultimately be fruitful for the organization. For example, Cosentino and Paoloni (2021) show that female managers have played a more effective role than men in crisis management during the digital transformation and not only did they facilitate the management of the transition, but also established strong formal relationships with institutional stakeholders. However, men have more opportunities in organizations, and this has prevented women from learning a sort of skills (Gudjonsson et al. 2022). In fact, there is limited knowledge about empowering women in organizations, which has hindered the development of women in the organization (Al Hakim et al. 2022). The current study aims to investigate the role of mentoring in balancing women's work-life balance. Mentoring, indeed, is one of the training and development approaches of employees. Mentoring is a process in which a capable and knowledgeable person advises a person with less experience to learn, grow and develop in their profession and specialty (Allocco et al. 2022). Mentors can help women cross social networks and build social capital across gender boundaries (Reis and Grady 2020). Mentoring plays an important role in the personal and professional development of employees and prevents barriers and stereotypes that prevent women from accessing better positions in the organizational hierarchy. One of the consequences of individuals' participation in mentoring programs is to reduce the conflict between the multiple roles of individuals and to balance the work and family roles of women.

Many female managers and professionals face countless challenges in the workplace and even within their own families. They need to balance their work and personal lives by relying on informed choices, because it can easily overshadow their physical and mental health. In this regard, organizations providing services to citizens, including the municipality, are no exception to these rules, and managers and officials are involved in such issues. Organizations, especially those that are responsible for sensitive and critical jobs, need human resources loyal to the values and goals of the organization. The municipality is a legal, local, and independent organization that has been formed within the city to meet the development, welfare, and service needs of city residents at the local level. It is one of the largest leading organizations with a high degree of complexity and expertise. On the other hand, due to the nature of municipal activities, which is one of the most important and fundamental organizations in order to provide services to citizens, it is necessary to conduct this research in this statistical community. It seems that designing an appropriate model of work-life balance in the context of a mentoring program and its proper implementation represents one of the important factors that can affect the process of implementing strategies and decision-making in the coming years of organizations.

According to the Iranian Statistics Center, the total population of working-age women in Iran in 2022 is estimated at 31,783,000, of which only 3,681,000 (i.e., 11.6%) have jobs (Iranian Statistics Center 2022). On the other hand, the share of women in employment is equal to 17.3% (and the share of men is 82.7%) (World Bank 2021). Although there is no study in the literature that examines the impact of culture and the extent of men's influence on women's employment in Iran, this difference and discrimination in women's employment can be attributed to the male dominant culture of this society. It is much more difficult for women to promote their careers and reach managerial and high organizational positions, perhaps due to cultural reasons. Tehran Municipality, due to its mission and organizational culture, needs a model to implement its training and development to achieve its goals. Women empowerment is one of the goals of Tehran Municipality, therefore, Tehran Municipality is one of the first organization that implemented the female human resource development program through mentoring programs. Thus, it is necessary to identify the role of training and development of human resources (i.e., mentoring) in women work-life balance and to improve the strategic capabilities of women. If the organization fails to develop and implement a favorable process for the work-life balance model for women, in addition to wasting their developed talents, in the future, there will be a lack of suitable and specialized people for important key positions in the organization. The lack of a proper and

localized model of work-life balance in mentoring in the organization makes the efforts to make the best use of the elites and talents unorganized and faces numerous ambiguities in the implementation path; it also poses a serious challenge to the goals set in the use of talent and ultimately the realization of organizational strategies. The research objectives of this article are: (a) providing a suitable model of work-life balance of women in the mentoring programs; (b) understanding the relationship between the work-life balance of women and the mentoring programs, which can play an important role in human resource training and development programs and in improving the career path of individuals, especially women in organizations; (c) creating the right mentoring process in the workplace to give women the ability and power to have a more integrated perspective and less conflict over their different roles in the organization, family, and community; and (d) in organizations and institutions such as municipalities, there is a need to provide solutions to solve problems that exist in the field of managing women's roles and performing their duties, e.g., more job and family satisfaction, providing better and more satisfactory services to the citizens in the fastest possible time and with appropriate quality.

In the existing literature, there is no study that presents how mentoring affects women's work-life balance. Therefore, the findings of the present study can help Tehran Municipality in designing comprehensive mentoring programs in order to create a work-life balance for women. That is why the present study seeks to model the mentoring of work-life balance of women in Tehran Municipality. Based on the explanation of the problem and the importance of the research, the main questions of the present research are:

- What are the dimensions and components of women's work-life balance in mentoring?
- What is the work-life balance model of women in mentoring?

## 2. Theoretical Foundations and Research Background

### 2.1. Mentoring

Mentoring is one of the strategies that has been used in recent decades to develop and improve human resources in the organization, and it is the process of an informal transfer of knowledge, social capital, psychological and social support of mentors to the causes of their career advancement (Read et al. 2020). Mentoring involves a direct, long-term relationship between an experienced and knowledgeable person and an inexperienced person who strives for professional, educational, and personal training and development (Sandardos and Chambers 2019). Mentoring with personal support for individuals improves social well-being, increases self-esteem, and creates a sense of accomplishment in the mentees. In contrast, mentoring job support for mentees increases job satisfaction in terms of career advancement, salary increase, etc. (Salem and Lakhal 2018).

There are two main categories of mentoring process functions in literature: career functions and psychological functions. Career functions refer to the mentor position and influence in an organization. In contrast, psychological functions refer to the personal connection between the mentor and the mentee that is established at the beginning of the relationship (Istenič Starčič and Kovač 2009). Throughout every stage of life, a person receives support from family, friends and peers that leads to growth and development. Individuals are also supported by a network of advocates on how to adapt to the social world and career advancement. On the other hand, during each stage, the individual learns and grows professionally and personally by modeling the experiences, attitudes, and values of individuals. Mentors create positive personal value and self-confidence in mentees by providing acceptance and approval tasks (Stockkamp and Godshalk 2022). In addition to maintaining professional working relationships, including mentoring relationships, individuals also establish personal relationships with family, friends, and peers. Accordingly, for the theoretical framework applied in this study, both functions of mentoring will be used. These functions provide a context for understanding the experiences of participants in this research.

### 2.2. Work-Life Balance

Having different roles in the workplace has kept people away from other aspects of life (Sarker et al. 2021). However, today's lifestyle has created contradictions in relation to the two categories (i.e., work life and personal life). One of the most important requirements for people to have a good job performance is their ability to meet the multiple and conflicting expectations that are expected of a person who is in different roles at the same time. One of the main roles that people play is their simultaneous role in the organization and family. Family and work are two important aspects of each person's life, and coordination and proportion between these two aspects affect the overall health of people. The need to play an effective role in these two areas has been raised in a concept called work-life balance. Problems related to work environments affect work-life relationships, and these types of role-playing conflicts are one of the most important sources of stress (Butt et al. 2022; Kooli 2021; Maqsood et al. 2021). The incompatibility between the requirements related to work and life roles has created many problems for individuals to meet the needs in both areas. Reducing the conflict between work and life will increase employee satisfaction in the areas of work and personal life, as well as affect job outcomes. Paying attention to issues related to work-life balance and employees' personal lives can be used as a mechanism to help retain employees (Weale et al. 2020). Work-life balance is employees' perception of work and non-work activities that are in line with current life priorities and lead to their growth and promotion in the work (Gragnano et al. 2020). Due to the importance of the topic, many studies have been conducted to investigate various aspects of creating a work-life balance, especially for women. For example, Kooli (2022) examines the strengths and weaknesses of different work standards, such as remote working, in creating a work-life balance for women. He states that although remote working can bring workplace flexibility and the opportunity to work from the comfortable home for women, it causes the integration of roles and not only creates distractions to perform their duties. In addition, it affects their time family and private space. Work-life balance occurs in workspaces when limited resources are evenly distributed between work and personal life. Balance in resources does not mean equality, but rather the proper allocation of resources based on goals and abilities. According to the issues raised, work-life balance is a strategic tool for human resource management and a key element in human resource maintenance strategies, especially for women.

### 2.3. Work-Life Balance and Mentoring Process

The concepts of work and personal life of individuals used to be considered as two separate priorities. However, in order to increase the performance (Beauregard and Henry 2009; Lazar et al. 2010) and satisfaction (Abendroth and Dulk 2011; Haar et al. 2014; Rani and Mariappan 2011) of the employees, organizations are continuously looking for solutions through which they can create a balance between the work and life of their employees. Employees who spend all day working or working long hours in the workplace face the challenge of balancing their personal lives with job demands (Odengo and Kiiru 2019). The constructive balance between work and personal life is very important for working women, especially in the current situation where the family environment and workspaces have created many challenges and problems for women (Alqahtani 2020). Because most family responsibilities are usually borne by women, women generally experience lower levels of work-life balance.

Mentoring brings potential benefits to women by focusing on their professional and personal development. Given the above, the results of existing research show that both mentoring and the balance between women's work and personal life are important work attitudes that helps keep women in the workplace. By managing women's work-life balance, the managers of the organization can have happy, satisfied, and lively employees who can demonstrate effective performance in their work and personal life with interest, loyalty, and commitment to the organization. The following is a summary of related research in Table 1.

**Table 1.** Notable literature on mentoring and work-life balance.

| Researchers | Title | Results |
|---|---|---|
| (Asghar et al. 2018) | The impact of work-family conflict on turnover intentions: the moderating role of perceived family supportive supervisor behavior | Mentoring reduces the conflict between the professional and personal lives of individuals. |
| (DeMeyer and DeMeyer 2018) | Mentoring the next generation of authors. | Good mentoring balances the work-life of individuals and also increases motivation, skills development, knowledge and contributes to the professional ethics of employees. |
| (Adnan Bataineh 2019) | Impact of work-life balance, happiness at work, on employee performance. | By implementing flexible work plans, caring for children of working mothers, parental leave, and supervisor support for employees, organizations reduce the conflict between individuals' personal and professional lives, increase job satisfaction, reduce job stress, and reduce employee intentions. |

A review of existing studies in the literature shows that the main focus of existing studies has been on the impact of mentoring on increasing job satisfaction, self-confidence, organizational commitment, and self-management (Banerjee-Batist and Reio 2016), using mentoring for teacher training and improvement (Wang and Bale 2019), implementation of mentoring process for professors and doctoral students and faculty members of the university (Espino and Zambrana 2019), and the effect of mentoring on education and clinical competence of nursing students and their intention to resign (Clochesy et al. 2019; DeMeyer and DeMeyer 2018). A close look at these studies shows that, first, according to the results of studies, attention to the issue of work-life balance of women at all organizational levels is one of the essential needs in the present era. Second, according to the literature, no study was conducted directly on the issues of work-life balance and women in mentoring in municipalities. Most of these studies are related to mentoring and training men.

According to the literature review, most of the research that has been done so far in the field of work-life balance and mentoring has proposed models that are based on quantitative and survey research methods and there is no process-comprehensive model of work-life balance in mentoring in the research. These studies revealed that in the design of mentoring programs, in addition to mentees, benefits for the mentor should be also considered. Besides, by carefully examining mentoring and its output, effective steps can be taken in designing mentoring.

## 3. Research Methodology

This is a qualitative developmental applied study, because it uses constructivist grounded theory recommended by Charmaz (2014) to design a model of the impact of mentoring on work-life balance for women, which is currently unparalleled, it is developmental, and since it takes place in response to a problem, it is applied research. Researchers' studies in all fields are designed and implemented based on basic philosophical principles, which appear in their activities and research stages. The roots of these philosophical principles are in three concepts of ontology, epistemology, and methodology, which are considered as philosophical foundations (McNabb 2015). The present study is a grounded theory study based on the interpretive paradigm while ontology is based on relativism. Relativism means that reality is a mental thing and varies from person to person. In the present study, the epistemological assumptions of the research are based on constructivism and the researcher seeks to create a model by analyzing the interview findings to construct the model. In addition, the methodology reviews, collects, and analyzes research data.

Interpretive methodology is based on understanding the phenomena from the perspective of the subjects. Interpretive theory is usually grounded theory. That is, it arises from the data, not precedes it.

Given that the present study designs a model for the effectiveness of mentoring on the women's work-life balance, the research approach is inductive. The inductive approach draws a general conclusion from a particular sample and, basically, scientific and research advances are made using induction. On the other hand, due to the theoretical gap that exists in the impact of mentoring on women's work-life balance, in the present study, the grounded theory has been used. As a result, the strategy of the present study is to create the theory based on the data. Cross-sectional studies for many studies are conducted in such a way that relevant data are collected only once, e.g., over several days, weeks, or months, to answer research questions. Therefore, this is a cross-sectional study due to the data collection method.

### 3.1. Sampling

The present study population consists of managers, deputies and specialists participating in mentoring in different district of Tehran Municipality. Tehran Municipality is one of the pioneers of using mentoring programs for the development of human resources. Nevertheless, the mentoring programs are in the initial phase of implementation, and therefore this program is designed only for the high-level employees of the organization, and it is expected that the mentoring program will be expanded for all members of the organization at different organizational levels in the future. For sampling, theoretical, purposeful sampling methods are used, and the samples are selected for the purpose of the research. The sample size is determined during the research. Data analysis was performed simultaneously with data collection. The selected participants of this study include 20 deputies, managers and specialists participating in mentoring as mentors and mentees, who are working in Tehran Municipality and have at least 10 years of relevant work experience. Further, 15 of 20 participants were female and 5 of them were men. Seventeen of the participants were mentees and only three of them were mentors. All five men were married, while eighty percent of the women participating in this study (i.e., 12 of them) were married. It should be noted that since the purpose of the article is to investigate the impact of mentoring on women's work-life balance, only female mentees were interviewed, at first. However, after consulting with university professors, to increase the validity of the findings, mentors and male mentees were also considered to be interviewed and asked about women's work-life balance. Table A1 in the Appendix A shows the demographic characteristics of the interviewees in terms of gender, level of education, work experience, type of participation in mentoring, and the duration of the interview.

### 3.2. Data Collection Method

To study the literature and research background, the method of studying existing libraries and articles has been used. The method of constructivist grounded theory was used to identify the subject under study. For this purpose, detailed and semi-structured interviews were conducted with deputies, managers, and specialists participating in mentoring as mentors and mentees, who are working in Tehran Municipality and have at least 10 years of relevant work experience. It should be explained that the interviews were conducted during a four-month course from mid-April 2022 to mid-July 2022. Examples of questions asked in interviews are:

- How is a constructive and effective relationship between the mentor and the female mentee formed in mentoring?
- Is the gender of the mentor effective in forming the relationship between the parties? How?
- Do mentors help balance women's work and personal roles?
- What are the factors that improve the balance between women's work and personal plans in the relationship between the mentor and the mentee?

- Can the friendship relationship between the mentor and the female mentee help manage women's work and personal roles?

*3.3. Data Analysis Method*

The first step in data analysis in grounded theory is coding. The codes of each interview describe how people pay attention to the subject. Data analysis was performed simultaneously with data collection, unlike quantitative methods. In addition, among the qualitative software, Maxqda software was used to analyze the research data. The process of data coding in the constructivist grounded theory method is performed in four stages of initial coding, focused coding, axial coding, and theoretical coding. Coding is an operation in which data are parsed, conceptualized, and put together in a new form (Glaser 2007). In this study, four stages of coding were performed in the continuous interaction between the researcher and the interviewees.

Initial coding: Open coding is the communication process between the interviewee and the interviewers to gather the data needed for the study. In this coding, a title related to the content of the interviews is assigned to them.

Focused coding: in focused coding, the titles extracted from the initial coding stage are categorized and divided into separate sections and evaluated in more detail to distinguish their similarities and differences. A label is also assigned as a concept for each of these categories.

Axial coding: The third step in data coding in constructivist grounded theory is axial coding. The main purpose of this type of coding is to identify and examine the categories. At this stage of coding, the theory gradually becomes apparent and the relationship between the categories is defined.

Theoretical coding: The last step in the coding process in constructivist grounded theory is theoretical coding, through which the relations between the derived categories are determined. In this research, the data are analyzed, and reasons based on the texts, comments of experts, as well as field observations are presented. The product of this process is the formation of a theoretical model

**4. Research Findings**

In the present study, in the initial coding stage, 1397 key phrases, in the focused coding stage, 227 concepts, and in the axial coding stage, 23 sub-categories were obtained. Finally, in the theoretical coding stage, all sub-categories were integrated into a main category. At the end of the analysis, the theoretical model resulting from the research coding shows the categories and relationships among the research elements. Table 2 shows examples of initial codes.

**Table 2.** Examples of initial codes.

| Initial Code | Interview Text | The Key Point |
|:---:|:---:|:---:|
| B10 | One of the important factors for the implementation of mentoring is to prepare the appropriate organizational context and infrastructure for the implementation and participation of employees in mentoring. | Providing organizational context and infrastructure |
| C17 | One of the main factors for implementing mentoring is paying attention to organizational maturity. In other words, if organizations have not reached an appropriate level of maturity, they cannot effectively implement mentoring. | Effectiveness of organizational maturity |

**Table 2.** *Cont.*

| Initial Code | Interview Text | The Key Point |
|:---:|:---:|:---:|
| E5 | One of the main approaches of the mentor in mentoring is to involve all the female mentees in each other's duties, that is, everyone can do each other's activities properly and there is no disruption in the activities. | Effective job rotation |

The second stage of coding is focused coding. Focused code appears more frequently between primary codes or is more important than other codes. Table 3 shows an example of how these codes are formed.

**Table 3.** Examples of focused codes.

| Focused Code | Key Concepts | Related Initial Codes |
|:---:|:---:|:---:|
| FF14 | Effective communication between the parties | F19-F38-F39 |
| FI47 | Providing the infrastructure to run the process | I74-I97-I103 |
| FG42 | Existence of mutual trust | G50-G62-G67 |

Axial coding is the third step in the grounded theory process, which is done to make connections between categories and subcategories. This process continues until theoretical saturation is achieved. In the present study, the latest new concept emerged in interview #11 and the emerging categories were completed to an acceptable level. Seven more interviews were then conducted to ensure theoretical adequacy. Due to the page limitation, examples of axial coding conducted in this study are presented in Table 4, and all axial codes resulting from the qualitative analysis of this study are presented in Table A2, in the Appendix A.

**Table 4.** Examples of Axial coding.

| Main Category | Subcategories | | Focused Codes |
|:---:|:---:|:---:|:---:|
| impact of mentoring process on women's work-life balance | Antecedents | Individual Considerations | Behavioral | Mental maturity |
| | | | Being responsible |
| | | | Being independent |
| | | Specialized | Trying to reach the goal |
| | | | Readiness to attend mentoring |
| | | | Mentoring acceptance |
| | | | Interest in participating in mentoring |
| | | | Feeling the need to participate in mentoring |
| | | | Female mentee's desire to learn |
| | Central Category | Role Management | Task Division |
| | | | Job Sharing |

The last step in the analysis process in the constructivist grounded theory is theoretical coding. In this stage, similar concepts are collected and formulated into categories, and how a single category relates to other categories is clarified. Figure 1 shows the relationships between the categories obtained in the axial coding step.

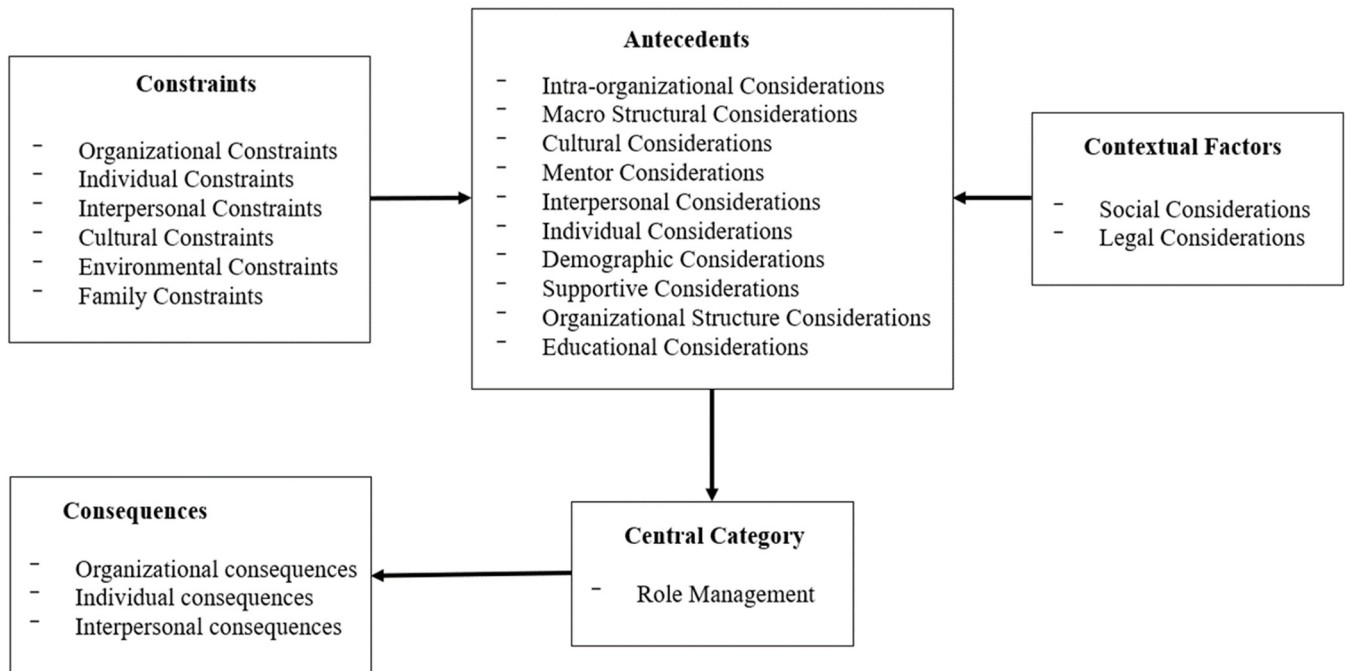

**Figure 1.** Framework of the impact of mentoring on women work-life balance.

*4.1. Central Category*

In this research, role management is considered as the central category. Because its traces exist in different parts of the data and play a pivotal role, in such a way that it can gather other categories around itself. Mentors familiarize female mentees with the work environment and future, strengthen the ability to think, make decisions and analyze independently in individuals, and encourage them to manage different roles. Division of labor is not only one of the most important tasks of a mentor, but also, based on that, mentoring achieves its set goals:

> " . . . *If the division of labor and tasks is balanced, that is, a combination of simple tasks and challenges to women, it can be effective. Because the parties trust each other, stress levels are reduced, and work efficiency is increased*". (Interview 3)

On the other hand, job sharing as a work planning method can be considered as a solution to the problems of women participating in mentoring. In this method, two or more people volunteer for a full-time job and share in all its benefits. Job sharing also occurs when two female mentees work together in the same way:

> " . . . *By sharing job responsibilities among female mentees, mentors increase expertise and efficiency of mentees when performing activities and leads to better management of organizational and personal tasks . . .* ". (Interview 13)

*4.2. Antecedents*

Antecedents lead to the creation and development of axial factors. Among the existing categories, cultural considerations, mentor considerations, individual considerations, organizational structural considerations, macro structural considerations, intra-organizational considerations, demographic considerations, supportive considerations, interpersonal considerations, and educational considerations are considered as antecedents. They play an active role in influencing mentoring on women's work-life balance, and as long as these considerations are not taken into account, the work-life balance of women in mentoring will not be achieved. Due to the great importance of culture in mentoring (Munday and Rowley 2022) and its remarkable impact on the career path and management of women's work and personal roles, there must be a proper culture of mentoring, its goals, and consequences in organizations.

" *. . . one of the important solutions is that the goals, characteristics, and consequences of mentoring are properly embedded in the organizational culture*". (Interview 1)

On the other hand, the mentor's professional and behavioral considerations are the mentor's professional and personality traits, which can create good interactions with the mentees as well as a good output for the organization. Individual considerations also refer to the same professional and behavioral characteristics of mentees, which causes professional and personal activities as well as mentee's interactions to be performed effectively. In order for mentoring to be implemented properly in organizations, considerations, such as creating a suitable foundation and infrastructure in the organization, must also be considered.

" *. . . In the work environment, according to the conditions, there must be the necessary and appropriate infrastructure for the effective implementation of mentoring*". (Interview 2)

In mentoring, policies and organizational structure must have the necessary flexibility that will make the process output more effective. Due to the complexities and environmental changes, the need for organizations to be flexible in organizational processes as well as the need for mentoring in organizations has increased. Mentoring as one of the most effective training strategies should be supported by organizations. Mentoring takes a relatively long time to achieve the desired results. In addition, the more the mentoring meets the needs of the organization, the higher the success of effective results. In addition, hiring in-house expert mentors can help develop and manage the roles of the mentees.

"*If mentees, especially women, feel safe and secure in their work environments, they can enter their home and personal life with a good mood due to the effective output they had in the workplace, and transfer this good feeling to their personal life as well*". (Interview 12)

The type of duties and responsibilities that the mentor delegates to the female mentees can be effective in managing work and personal roles. Mentors should plan careers for women with more responsibilities in their personal lives so that the parties are familiar with each other's responsibilities and interact with each other. If the job descriptions are well written, it will provide a clear picture of the job and facilitate the process of performing tasks.

"*In order for mentoring to be effective in better managing the role and personal roles of female mentees, the precise definition of the duties and responsibilities of the mentees must be provided*". (Interview 6)

Paying attention to the provision of workplace wellbeing in retaining efficient female employees is known as a strategic factor. Having flexible work plans helps female mentees to better coordinate the needs of their family and organizational life. It is better for women to have fewer working hours in the organization due to more tasks. In addition, mentors should support them in using different leave as needed.

"*Since we participated in mentoring, our mentor has planned the tasks in such a way that we can use our vacations at special intervals, and this has improved the morale and job satisfaction of the female mentees in the organization*". (Interview 5)

The cooperation of family members reduces the heavy burden of responsibilities on an individual. Cooperation among family members also helps to increase mutual understanding and empathy of members. The physical and mental comfort of women in the family is established in the work environment and society.

"*A female mentee should have peace of mind both in the family and in the workplace. If a female mentee is not calm in the family environment and is constantly involved in many problems, she will certainly bring stress and tension into the workplace*". (Interview 15)

The mentor's emotional support should be commensurate with the needs and expectations of the mentees. Friendship with a mentor can help to improve the relationship between the parties and increase the spirit of the female mentee, as well as create positive

consequences. Therefore, establishing the right relationship and mutual understanding can play a decisive role in the atmosphere of the organization. To improve the relationship, the parties should be similar in terms of personality traits and thinking style. The success of the parties in the relationship also depends on the success in achieving the set goals.

> *"In the process of mentoring, in order to improve the relationship between the mentor and the female mentee, it is better for them to empathize with each other, think together and think of common goals so that they can have a suitable output for themselves and the organization"*. (Interview 4)

When the parties trust each other, information and resources are more easily shared, and cooperation is more effective. Men, on the other hand, need to learn proper management practices in relation to women participating in mentoring. In addition, mentoring training programs are planned to improve the level of effectiveness and familiarity of employees with the goals and contents. It will be more effective if the flexibility in holding mentoring training programs is increased and newer methods are used.

> *"It is important mentoring process training programs have the necessary flexibility. In other words, each organization should have program design and content development according to its organizational conditions"*. (Interview 2)

*4.3. Contextual Factors*

Contextual factors play a role in the impact of mentoring on women's work-life balance. Contextual factors include codes that specifically affect the antecedents in the work-life balance of women in mentoring. The contextual factors include two sub-categories, which are: social considerations and legal considerations. The social support perceived by women is recognized as one of the important social factors affecting the improvement of their morale, productivity, and effectiveness and can have a positive and direct impact on the quality of their working and personal lives. On the other hand, given the discrimination and challenges imposed on women's society, legislators should strive to enact laws and policies to prevent the spread of stereotypes and discrimination and to effectively promote women's rights.

> *"If the laws and policies of society are revised and amended, a number of discriminations against women may be eliminated"*. (Interview 11)

*4.4. Constraints*

Constraints are factors that adjust the influence of mentoring process on women's work-life balance. Findings reveal that the following factors are considered as the barriers constraining the effect of mentoring process on women's work-life balance: organizational constraints, individual constraints, interpersonal constraints, cultural constraints, environmental constraints, and family constraints. Behavioral patterns, cultural restrictions on women in society and organizations, has limited the active participation of women in mentoring. Meanwhile, the capabilities of women in various fields have been proven many times. Women's self-confidence is also essential in any situation and can have a tremendous impact on their personal and professional lives. Unfortunately, the cultural issues have prevented women from growing as well as men in the society, and this has reduced their self-confidence. On the other hand, " . . . *women in organizational environments have also faced mental and emotional conflicts and challenges due to facing some limitations and challenges*" (Interview 1).

It is crystal clear that women are physically and mentally different from men. Traditional workspaces restrict women. They may be confronted with a number of organizational methods and policies that may not be able to develop and effectively manage their roles according to their competencies. "One of the constraints in creating flexible organizational structures and processes is the existence of traditional perspective and atmosphere that exist in the administrative system of public and semi-public organizations . . . " (Interview 6). The position of women in organizations is not defined as equal to men: job security as

well as their salaries are lower than men, and excuses, such as women's inexperience in crisis management, are raised.

> "*Most organizational decision-makers are men, they are closer to the top officials of the organization, and they decide about the number of female managers. Their perception is that most of their mental concerns are with family responsibilities, and they may not be able to spend adequate time on organizational tasks properly. Certainly, these things affect their career path*". (Interview 12)

Since in most public and semi-public organizations, organizational structures and policies are relatively inflexible, it is to some extent to the detriment of women, and it creates many difficulties in women's careers. As a result, women do not have enough peace of mind, and this may lead to a work-life imbalance. A female mentee may also perform various roles in the organization due to her competencies. The multiplicity of organizational roles can have a detrimental effect on the effectiveness of women's activities. Organizations still do not accept mentoring as one of the training and human resource development approaches. Organizations need specialized mentors for effective mentoring. "*One of the problems faced by public and semi-public organizations is the lack of specialized personnel in the role of mentor who can be actively involved in this process.*" (Interview 15).

Women, especially in Iranian society, will not be able to properly reach the stage of self-fulfillment of their talents and competencies. If proper organizational support for mentoring is not also provided, it can lead to negative consequences. On the other hand, women who do not have the skills to communicate properly with their mentor, this creates challenges for both parties and the organization. Due to the problems of working life, women are also faced with problems in their personal lives, which leads to the lack of proper management of their work and personal roles. "*Factors such as: family life responsibilities, lack of support from the family, number of children,* etc. *create challenges in the personal and work lives of female mentees*" (Interview 11). It is possible that the number of children also has negative effects on the management of women's job and personal roles. These multiple roles require multiple factors, such as time, attention, concentration, and adequate family support.

*4.5. Consequences*

Mentoring process improves female mentee's motivation and performance and increases their awareness of the organization so that they consider themselves part of the organization. As a result, the free flow of information and organizational transparency enhances the level of team trust between the parties. "*If mentoring is implemented as one of the training and human resource development approaches in organizations, of course organizational transparency will improve*" (Interview 6).

The presence of women in mentoring and their constructive interactions with mentors improves their organizational citizenship behaviors. "*For effective mentoring in the organization, it is better to make the right decisions about the financial support for the process and the staff involved in the process. Because when women involved in the process believe that they receive the right financial support, they act with heartfelt desire, and consequently their citizenship behaviors also improve*" (Interview 12).

Mentors familiarize women with the work environment, strengthen the ability to think and make decisions and analyze independently in individuals, and encourage them to strike a balance between personal life and career. One of the consequences of women's participation in mentoring is improved self-esteem. One of the needs of women in participating in mentoring is to achieve intellectual independence. In addition, their physical and mental capacity increases compared to the past which promotes their creativity. "*The necessary organizational support should be provided to female mentees, and they should believe that the system supports them. As a result, this will improve their motivation and creativity*" (Interview 2).

The better the mentor provides for the female mentee, the more relaxed she experiences and the better her physical health. Women's active participation in mentoring improves their quality of life "*The constructive support that the mentor gives to the female mentee makes*

*women happier, as well as having a higher spirit in their personal life. Of course, both their job performance and their quality of personal life improve.*" (Interview 5).

The presence of women in mentoring improves their performance. Mentors offer mentees ways to achieve career advancement and increase their ability to perform tasks. Improving the decision-making skills of women in personal and business matters is, in fact, about the right choice based on the right criteria. One of the factors affecting individual and organizational performance and efficiency is the presence of people in mentoring and cooperation between the mentor and the mentee. When people work together, they each have the opportunity to present their best ideas and efforts, ultimately creating creativity and increasing the productivity of the parties to the relationship. Mentors and mentees understand the importance of good communication. If constructive interactions are established between the parties, this will improve mutual understanding over time.

*4.6. Theoretical Coding*

In the theoretical coding stage, the relationship between the criteria of the model of the effect of mentoring on the work-life balance of women was determined in the form of research narrative analysis. Based on the theoretical coding, the following statements can be proposed:

- Antecedents, including cultural considerations, mentor's considerations, individual considerations, organizational structural considerations, macro structural considerations, intra-organizational considerations, supportive considerations, demographic considerations, interpersonal considerations, and educational considerations, affect role management (as the central category).
- Contextual factors, including social considerations and legal considerations, affect role management through antecedents.
- Constraints, including organizational constraints, individual constraints, interpersonal constraints, cultural constraints, environmental constraints, and family constraints, affect role management through antecedents.
- Roles management is achieved as a central factor based on antecedents, taking into account social and legal considerations and constraints. Role management will result in the realization of individual, organizational, interpersonal, and family consequences.

## 5. Conclusions and Recommendations

Due to the many benefits, it can have, the necessity of gender diversity in the workplace has been emphasized in the literature (Fine et al. 2020; Mousa et al. 2020). However, there is limited knowledge about the empowerment and development of women in organizations, which has prevented women from equal development opportunities (Al Hakim et al. 2022). Especially when women have multiple, and sometimes contradictory, responsibilities in the work environment and the family environment, creating a balance between work and life roles is very necessary. Therefore, the present study tried to investigate the role of mentoring in balancing women's work-life as a tool for empowering women. This study disclosed that the active participation of women in mentoring can have positive consequences for both women and organizations. There are considerations (which were labeled as antecedents in this study) that organizations should consider for proper implementation of mentoring and improvement of results. The support systems provided by managers and mentors increase the well-being and safety of female mentees. Mentor support is provided in the form of professional and behavioral support to female mentees and leads to higher job performance, increased job and organizational commitment, and better communication between the parties. These findings are consistent with the results of the research of Sar et al. (2017). Therefore, mentor behavioral support for female mentees creates a sense of success in them. In contrast, mentor's professional support increases the job satisfaction of female mentees, and these results are consistent with the findings of Salem and Lakhal (2018).

When mentees trust mentors, they share their thoughts, feelings, and experiences with the mentors. This trust can lead to significant efforts to get to know each other. These results

are in line with the findings of the study by Hieker and Rushby (2020). It is also revealed that the active participation of women in the mentoring can help their career advancement in the organization. These results are in line with the findings of Reis and Grady (2020).

Due to the balance in work and personal roles, women are generally more satisfied with their job, which can lead to a more positive work environment and a higher level of job desire. In addition, mentor behavioral support for female mentees improves their sense of sociality, self-esteem, and leads to a greater sense of accomplishment. In contrast, mentor's professional support for female mentees throughout their careers increases job satisfaction in terms of career advancement and increases salaries and benefits, and these results are consistent with the findings of Chatterjee et al. (2021).

Mentoring deepens the mentees' familiarity with the atmosphere and future of their work ahead, strengthens their ability to think, make decisions and analyze independently, and encourages them to strike a balance between their personal lives and their careers. In fact, mentors provide the needs of the mentees in a timely manner and are effective in the personal and professional development of the mentors. These results confirm the findings of the studies of Reis and Grady (2020).

On the other hand, the importance of modeling the role of women as mentors in mentoring can be used as an important factor to address challenges in the workplace such as: discrimination, gender stereotypes, and the management of women's roles. However, the findings of the present study confirm that transgender mentoring relationships are effective in the relationship between women because women with the support of male mentors can have broader, more diverse views and learn different behaviors. These results are in line with the findings of the study by Reis and Grady (2020).

The contribution of this study is the modeling of the processes of mentoring's impact on women's work-life balance using constructive grounded theory. The findings of this study, indeed, provide guidelines including effective mechanisms, constrains, contextual factors, and expected consequences of the impact of mentoring on creating a work-life balance for women, which can be used by Tehran Municipality to design mentoring programs effectively. The presented theoretical framework can be placed as a foundation for the study of future researchers as well as similar service organizations. A cross-sectional study, for example, is recommended to provide empirical evidence of testing quantitatively the theorical framework proposed in this study. Besides, a potential longitudinal study can be performed before and after employee participation in the formal mentoring. Further research should be conducted on the perspectives of men and women participating in mentoring and their experiences of managing work and personal life. It is also recommended to study appropriate organizational structures for the effective implementation of mentoring in different organizations, according to the specific conditions and characteristics of organizations and their human resources.

However, in order to generalize the findings of the present study, there are limitations that should be considered in future studies. First of all, since the number of articles published in this field was very few, the discussion on the existing literature and the identification of different aspects of the effects of mentoring on the work-life balance were limited to the articles that were reviewed in this article. Second, given that the research has been conducted in the semi-public sector and given the political pressures that we often see in the public and semi-public sectors, there is a problem in that a number of interviewees are cautious and are very careful in answering the questions, which can affect the results of the research. Finally, the findings of the present study are limited to the period of data collection and by changing the conditions and time, the results of the research can certainly be changed. Due to these limitations, future researchers are recommended to compare the work-life balance of women in mentoring in the governmental and semi-governmental sectors of different countries to model successful experiences. It is also recommended that a study be conducted to examine the extent to which women in the workplace have access to mentoring and how a culture of mentoring can provide more opportunities for women.

**Author Contributions:** Conceptualization, P.B. and K.P.; methodology, P.B. and S.N.; software, P.B.; formal analysis, S.H.; investigation, P.B. and K.P.; data curation, S.N. and S.H.; writing—original draft preparation, S.N.; writing—review and editing, S.H. All authors have read and agreed to the published version of the manuscript.

**Funding:** This research received no external funding.

**Institutional Review Board Statement:** Not applicable.

**Informed Consent Statement:** Participation in this study was completely optional and has been done with the full consent of the participants (who are municipal employees). Besides this, the participants have been assured that all the collected information will remain anonymous, and this information will be used only for the purpose of conducting this study, i.e., studying the impact of mentoring processes on balancing women's work-life. And the collected data will not be shared with any person or entity.

**Data Availability Statement:** Not applicable.

**Conflicts of Interest:** The authors declare no conflict of interest.

## Appendix A

**Table A1.** Profiles of the participants.

| Participant Code | Job Title | Gender | | Education | Work Experience | | Type of Participation | | Interview Duration | |
|---|---|---|---|---|---|---|---|---|---|---|
| | | Female | Male | | Over 15 Years | Less Than 15 Years | Mentee | Mentor | More Than 1 h | Less Than 1 h |
| A | Deputy Head of Culture and Urban Development | | ✓ | Masters | ✓ | | ✓ | | | ✓ |
| B | Master of Human Resources | ✓ | | Masters | | ✓ | ✓ | | | ✓ |
| C | Director of Veterans Affairs | | ✓ | Masters | ✓ | | ✓ | | ✓ | |
| D | Finance and Administration Manager | ✓ | | Masters | ✓ | | ✓ | | | ✓ |
| E | Archive Manager | ✓ | | Bachelors | ✓ | | ✓ | | ✓ | |
| F | Head of Cultural and Social Affairs | ✓ | | PhD | ✓ | | | ✓ | ✓ | |
| G | IT Manager | ✓ | | Masters | ✓ | | ✓ | | | ✓ |
| H | Human resources expert | ✓ | | Masters | | ✓ | ✓ | | | ✓ |
| I | Human Resources Manager | ✓ | | PhD | | ✓ | ✓ | | ✓ | |
| J | Head of Cultural Affairs | ✓ | | Masters | ✓ | | ✓ | | ✓ | |
| K | Health Manager | ✓ | | PhD | ✓ | | | ✓ | | ✓ |
| L | Master of Human Resources | ✓ | | Masters | ✓ | | ✓ | | ✓ | |
| M | Municipal Services Manager | | ✓ | Masters | ✓ | | ✓ | | | ✓ |
| N | Director of Administrative Affairs | ✓ | | Masters | | ✓ | ✓ | | | ✓ |
| O | Head of Welfare Affairs | | ✓ | Masters | ✓ | | ✓ | | | ✓ |
| P | Financial manager | ✓ | | Masters | ✓ | | ✓ | | | ✓ |
| Q | Director of Administrative and Recruitment Affairs | | ✓ | Masters | ✓ | | ✓ | | | ✓ |
| R | Director of Education | ✓ | | Masters | | ✓ | ✓ | | | ✓ |
| S | Master of Human Resources | ✓ | | PhD | ✓ | | ✓ | | ✓ | |
| T | Human Resources Manager | ✓ | | Masters | | ✓ | | ✓ | | ✓ |

**Table A2.** Axial coding.

| Main Category | Subcategories | | Focused Codes |
|---|---|---|---|
| Dimensions and components of the impact of mentoring on the work-life balance of women | Antecedents | Cultural Considerations | Making mentoring as a culture |
| | | | Cultural support for women |
| | | Mentor Considerations | Behavioral |
| | | | being honest |
| | | | Being kind |
| | | | Being an effective listener |
| | | | Justice in interactions |
| | | | Flexibility |
| | | | Cultural Intelligence |
| | | | Emotional Intelligence |
| | | | Role modeling |
| | | | Effective expression power |
| | | Specialized | Being an expert |
| | | | Availability |
| | | | Problem solving ability |
| | | | Effective leadership skills |
| | | | Decision-making skill |
| | | | Long-term vision |
| | | | Awareness of women's psychology |
| | | Individual Considerations | Behavioral |
| | | | Mental maturity |
| | | | Being responsible |
| | | | Being independent |
| | | Specialized | Trying to reach the goal |
| | | | Readiness to attend mentoring |
| | | | Mentoring acceptance |
| | | | Interest in participating in mentoring |
| | | | Feeling the need to participate in mentoring |
| | | | Female mentee's desire to learn |
| | | Organizational Structure Considerations | Internal Processes |
| | | | Infrastructure for mentoring |
| | | | Level of organizational maturity |
| | | | Flexible organizational policies |
| | | | Mentoring flexibility |
| | | | Effective mentoring targeting |
| | | Organizational Structure | Flexible organizational structure |
| | | Intra-organizational Considerations | Managerial Considerations |
| | | | Managers' support for mentoring |
| | | | Managers' interactions with women |
| | | | Supervising the training course |
| | | | Monitoring the implementation of mentoring |
| | | | Employing expert mentors |
| | | | Peace of mind in the workplace |
| | | | Long-term mentoring |
| | | | Organizational transparency |
| | | | Using internal mentors |
| | | | Mentoring needs assessment |
| | | | A systematic look at organizations |
| | | Macro Structural Considerations | Requires learning and flexible processes, mentoring |
| | | | Support for learning and flexible processes, mentoring |

**Table A2.** *Cont.*

| Main Category | | Subcategories | Focused Codes |
|---|---|---|---|
| Dimensions and components of the impact of mentoring on the work-life balance of women | Antecedents | Demographic Considerations | Gender of the mentor |
| | | | Age similarity of the parties |
| | | Job Supports | Type of job duties |
| | | | Career planning |
| | | | Career justice |
| | | | Flexible career path |
| | | | Job Description Definition |
| | | | Meaningfulness of job duties |
| | | | More diverse career paths |
| | | | Effective job rotation |
| | | Supportive Considerations — Motivational Supports | Welfare amenities |
| | | | Flexible work schedules |
| | | | Considering vacations |
| | | | Paying attention to motivational issues |
| | | | Flexible job tasks |
| | | | Considering breastfeeding hours |
| | | | Considering maternity leave |
| | | | Paying attention to telecommuting |
| | | | Presenting kindergarten |
| | | | Flexible working hours |
| | | Family Supports | Family support |
| | | | Mutual understanding of family members |
| | | | Interactions of family members |
| | | | Peace of mind in the family atmosphere |
| | | | Cooperation of family members |
| | | Financial Supports | Justice in payments |
| | | | Compensation for effective services |
| | | | Scientific indicators in the payment of benefits |
| | | Mentor Supports | Specialized support |
| | | | Behavioral support |
| | | Educational Considerations | Educating male managers to better understand women |
| | | | Holding a training course for mentoring |
| | | | Flexibility in the way of holding the training course |
| | | | Performance evaluation of the training course |
| | | | Holding a training course on maternity leave |
| | | Interpersonal Considerations | Constructive interactions of the parties |
| | | | Mutual respect |
| | | | Mutual understanding |
| | | | Mutual honesty |
| | | | Employing group mentoring |
| | | | Strategic thinking of the parties |
| | | | Long-term communication between the parties |
| | | | The human and moral view of the parties |
| | | | Intellectual similarity of the parties |
| | | | Having transgender vision |
| | | | Common goals |

**Table A2.** *Cont.*

| Main Category | Subcategories | | Focused Codes |
|---|---|---|---|
| | Antecedents | Interpersonal Considerations | Mutual trust |
| | | | Using virtual mentoring |
| | | | two-sided cooperation |
| | | | Collaborative decision making |
| | Contextual Factors | Social Considerations | Community supportive environment |
| | | Legal Considerations | Amending the general laws and policies of the society |
| | Central Category | Role Management | Task Division |
| | | | Job Sharing |
| | | Cultural Constraints | Restrictions and stereotypes against women |
| | | | Low self-esteem of women |
| | | Environmental Constraints — Legal restrictions | Job Restrictions for Women's Jobs |
| | | Environmental Constraints — Social constraints | The traditional nature of society |
| | | Individual Constraints | Physical limitations |
| | | | Mental conflicts |
| | | Interpersonal Constraints | Lack of mutual understanding |
| | | | Lack of constructive interactions |
| | | Family Constraints | Problems of family life |
| Dimensions and components of the impact of mentoring on the work-life balance of women | | | Number of children |
| | | | More responsibilities in the family |
| | | | Lack of family support |
| | Constraints | Organizational Constraints | Political behaviors in the organization |
| | | | Lack of culture of mentoring |
| | | | Lack of organizational support |
| | | | Traditional workspace of organizations |
| | | | Low flexibility in training and human resource development |
| | | | Low number of expert mentors |
| | | | Low number of female mentors |
| | | | Ignoring women in the workplace |
| | | | Lack of effectiveness of training courses |
| | | | Problems of multiple organizational roles |
| | | | Less job security for women |
| | | Management constraints | Lack of managerial support for women |
| | | | Lack of management support for mentoring |
| | | Financial limitations | Discrimination in payments |
| | | | Financial constraints of the organization |
| | | Job Restrictions | Barriers to women's careers |
| | | | Fewer career advancement opportunities for women |
| | | Structural constraints | Low level of organizational maturity |
| | | | Inflexibility of organizational structure |
| | Consequences | Organizational consequences | Improving Performance |
| | | | Reducing Costs |
| | | | Improving organizational culture |
| | | | Improving organizational citizenship behaviors |
| | | | Improve organizational commitment |

**Table A2.** *Cont.*

| Main Category | Subcategories | Focused Codes |
|---|---|---|
| Dimensions and components of the impact of mentoring on the work-life balance of women | Consequences | |
| | Interpersonal consequences | Improving mutual understanding |
| | | Mutual growth of the parties |
| | | Creating a sense of constructive competition |
| | | Improving mutual learning |
| | Individual consequences | Career advancement |
| | | Performance improvements |
| | | Increasing job satisfaction |
| | | Increasing job commitment |
| | | Reducing dropout |
| | | Individual development |
| | | Improving decision making |
| | | Improving sociability |
| | | Improving creativity |
| | | Improving physical health |
| | | Improving the quality of life |
| | | Improving self-confidence |

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
