# Peer review of "Modeling the Impact of Mentoring on Women’s Work-Life Balance: A Grounded Theory Approach"

_admsci, doi:10.3390/admsci13010006_

Round 1
Reviewer 1 Report
According to the literature, mentoring is one of the techniques or initiatives that can assist women in advancing their careers at the management level. Mentoring could help women overcome many barriers to breaking through the "glass ceiling," including work-life balance.
The purpose of this study is to create a model of work-life balance for women in the mentoring process (stated in line 204). However, there is some ambiguity in the writing regarding the study's objective.
It states that (Line 66) there is a need to identify the role of work-life balance in the mentoring process. (Line 218)- the mentoring process's impact on work-life balance. First, the author should provide additional clarification to explain the mentoring process itself. Mention the components of a mentoring programme, the process it refers to, and how work-life balance is incorporated into the mentoring process.
This study's respondents are those in positions of authority. Is there any junior or senior staff involved in the mentoring programme? While those in management positions have advanced their careers, we should also be concerned about female employees who are still in lower-level positions and hope that mentoring will help them advance faster. The author should explain why the manager or department head was chosen as the sample. How many women work for the company, and how many participate in the mentoring programme?
The highlight of this study is the work-life balance of women, however, there are male mentees in the sample. Why are they included? What role do they have in developing the model since the issue is about work-life balance for women employees?
The research methodology used in this study consists of several stages that result in a mentoring process model. However, no citations are provided to support the validity of the method.
(Line 248) The paper includes questions and interview text examples.
The paper should explain the basis of interview questions, as well as the scope or categories of questions asked, because asking the right questions will ensure the development of a comprehensive model. The model in Figure 1 is very detailed, but it is not reflected in the earlier steps of how each item in the model is built.
There was no statistical analysis to demonstrate that the elements discovered in this study (e.g., improved communication, trust, active participation in mentoring) could lead to job enhancement or reduce the imbalance of work and personal life. Because this model has yet to be empirically tested, some of the conclusions of this study are inaccurate.
The following items require revision:
1. Verify the sequence of tables 2, 3, and 4.
2. Under theoretical coding, Line 305 should be moved to Line 511.
3. The references list should be revised to remove duplicate names, page numbers, year of publication, and formatting.
Author Response
Dear Respected Reviewer,
First of all, we would like to thank you for reading our work and for providing valuable comments on our work. We agree with all your comments, and we strongly hold the belief that your comments help us to improve the quality of our work. Following you can find our response to your comments:
Your comment:
The purpose of this study is to create a model of work-life balance for women in the mentoring process (stated in line 204). However, there is some ambiguity in the writing regarding the study's objective.
It states that (Line 66) there is a need to identify the role of work-life balance in the mentoring process. (Line 218)- the mentoring process's impact on work-life balance. First, the author should provide additional clarification to explain the mentoring process itself. Mention the components of a mentoring programme, the process it refers to, and how work-life balance is incorporated into the mentoring process.
Our Response:
Thank you very much for this comment and we fully understand the point of your comment. We must say that the purpose of the article is to examine the impact of mentoring itself, not the processes and steps in a mentoring process. Therefore, and according to your comment, we replaced the word mentoring with the word mentoring process in the entire article to eliminate this ambiguity. However, in lines 39-42, a definition for the mentoring process is provided.
“Mentoring is a process in which a capable and knowledgeable person advises a person with less experience to learn, grow and develop in their profession and specialty.”
Regarding the objective of the article, the purpose of the article is clear, but as you mentioned, the choice of words and the structure of the sentences used to explain the purpose were not appropriate. Therefore, the sentences related to the purpose of the article were all rewritten and aligned with the story presented in the article. For example, lines 66-69, 98-99, were revised.
Your comment:
This study's respondents are those in positions of authority. Is there any junior or senior staff involved in the mentoring programme? While those in management positions have advanced their careers, we should also be concerned about female employees who are still in lower-level positions and hope that mentoring will help them advance faster. The author should explain why the manager or department head was chosen as the sample.
Our Response:
This comment is very relevant and correct, we also wanted to be able to cover all people at different levels of the organization who participate in mentoring programs. But since Tehran municipality is one of the pioneers of the mentoring program in organizations and since this program has been implemented recently, in the opinion of the organizational decision makers of the participants, only high-level positions were included in the mentoring program in the municipality. However, we have been told that this program is supposed to be expanded so that all municipal employees at all organizational levels can be included. Due to the importance of the issue and according to your comment, lines 272 to 276 have been added to the text of the article and the necessary explanations have been provided.
Your comment:
How many women work for the company, and how many participate in the mentoring programme?
Our Response:
The purpose of our article was to identify the theoretical framework of the impact of mentoring on women's work-life balance. Since the objective was NOT to investigate the performance of Tehran municipality as a case study, we have not collected data related to the total number of employees, the number of employees participating in mentoring, the number of women participating in Mentoring, the outputs and consequences of the implementation of mentoring in creating work-life balance of women in the municipality. We believe that this requires independent research.
Your comment:
The highlight of this study is the work-life balance of women, however, there are male mentees in the sample. Why are they included? What role do they have in developing the model since the issue is about work-life balance for women employees?
Our Response:
In fact, the authors of the article, with a same mindset, initially only interviewed female mentees. However, after consulting with university professors in order to confirm the validity of the findings, it was decided to take the expert opinion of mentors and also male mentees in this regard. Therefore, interviews were held with mentees and male mentors, who were asked about women's work-life balance. Subsequently, lines 247 to 254 were added to the text.
The purpose of this study, in fact, is to observe the theory that emerged from the interviews, so we tried to have diversity among the participants to have a more complete picture of the theory that is produced. That is why the participants were both married and single, men and women, mentors and mentees. For a cross-sectional study and when quantitative data is collected, only female mentors should be examined, and their marital status can be examined as an intervening variable. But in the study of grounded theory, the approach is different.
Your comment:
The research methodology used in this study consists of several stages that result in a mentoring process model. However, no citations are provided to support the validity of the method.
Our Response:
Thank you very much for your reminder. We used the constructive grounded theory introduced by Prof. Kathy Charms, in 2014, which was added to the text and the relevant reference was also added to the text.
Your comment:
(Line 248) The paper includes questions and interview text examples.
The paper should explain the basis of interview questions, as well as the scope or categories of questions asked, because asking the right questions will ensure the development of a comprehensive model. The model in Figure 1 is very detailed, but it is not reflected in the earlier steps of how each item in the model is built.
Our Response:
Since the approach of the article is the grounded theory, the authors should not have any preconceived notions or pre-determined ideas regarding what variables the research model should have in developing the conceptual model of the article. In other words, there were no targeted and predetermined questions to examine the various aspects of the subject under study. All efforts were made to extract a theory from the interviews. Nevertheless, in lines 264-272, an example of the questions was given, and also in Table 2, an example of the content of the interviews is given.
Your comment:
There was no statistical analysis to demonstrate that the elements discovered in this study (e.g., improved communication, trust, active participation in mentoring) could lead to job enhancement or reduce the imbalance of work and personal life. Because this model has yet to be empirically tested, some of the conclusions of this study are inaccurate.
Our Response:
We strongly agree with you that the theoretical framework presented in this article should be tested. Hence, nowhere in this article did we claim to have presented a conceptual model, but it has always been said that this article has extracted a theoretical framework from the interviews. Inspired by your comment, in the conclusion of the article in lines 586-587, we have suggested for future research to test the proposed theoretical framework of the article.
Your comment:
The following items require revision:
- Verify the sequence of tables 2, 3, and 4.
Corrected
- Under theoretical coding, Line 305 should be moved to Line 511.
I am afraid I could not understand which sentence it is, as theoretical coding started with lines 518-535, in the initial version of the article.
- The references list should be revised to remove duplicate names, page numbers, year of publication, and formatting.
Our Response:
Thank you for the comments. We tried our best to address all the comments.

Reviewer 2 Report
There is no doubt about the difficulties that women face in climbing the ladder of success in their working profession. Their tasks are typically hampered by a variety of factors such as family commitment, organisational commitment, culture, and many others.
The current study was carried out among selected Iranian women in top management positions in a Tehran municipality. Apart from the authors' efforts to model the impact of the mentoring process on Iranian women's work-life balance, the study's approach is intriguing given the fact that there is a lack of such a study in an Iranian woman.
I noticed some issues that should be addressed in the paper:
1. Introduction
Despite acknowledging the absence of such a study in Iran, the paper made no attempt to acknowledge Iranian culture's attitude toward allowing women to work.
a. A review of various Iranian studies on female labour force participation and employment in Iran reveals that, despite educational and social achievements, Iranian women's labor-force participation has not improved. Their participation rate remains low, their unemployment rate has risen in recent years, and their occupational options are limited. After Pakistan (3.7%), Iran was said to have the lowest percentage of women employed (9%).
b. Furthermore, Iranian husbands have the right to prevent their wives from working in certain occupations, and some positions require the husband's written consent. As a result, the authors' Introduction section would be more meaningful if they acknowledged the two issues raised above as the motivation for their research. It is only mentioned briefly in line 462.
2. Research Methodology
The study's sample consisted of Iranian women with at least 10 years of work experience who are managers, deputies, or specialists in a Tehran Municipality. This demonstrates that women hold top management positions in the aforementioned organisation.
a. The demographic information (Table A1) did not indicate whether the participants were married or single. Referring back to my point in 1.b. above, culture may have an impact on Iranian women's employment participation. According to some studies, married Iranian women are less likely to enter the labour force, whereas highly educated women, married or not, tend to stay at their jobs for longer periods of time.
b. The research should explain why these women in higher positions were chosen over those in lower positions. It would clear the air unless the current study intends to model a mentoring process specifically for women in top positions.
3. Conclusions
Based on the current approach, the study did not highlight apparent limitations that was not being incorporated in their study which is the inclusion of culture and the exclusion of Iranian women in lower positions in developing the model.
4. Typography errors
Rectify your Table numbering.
Line 294 – should be Table 3
Line 303 – should be Table 4
References:
see appendix
Author Response
Dear Respected Reviewer,
First of all, we would like to thank you for reading our work and for providing valuable comments on our work. We agree with all your comments, and we strongly hold the belief that your comments help us to improve the quality of our work. Following you can find our response to your comments:
- Introduction
Despite acknowledging the absence of such a study in Iran, the paper made no attempt to acknowledge Iranian culture's attitude toward allowing women to work.
- A review of various Iranian studies on female labour force participation and employment in Iran reveals that, despite educational and social achievements, Iranian women's labor-force participation has not improved. Their participation rate remains low, their unemployment rate has risen in recent years, and their occupational options are limited. After Pakistan (3.7%), Iran was said to have the lowest percentage of women employed (9%).
- Furthermore, Iranian husbands have the right to prevent their wives from working in certain occupations, and some positions require the husband's written consent. As a result, the authors' Introduction section would be more meaningful if they acknowledged the two issues raised above as the motivation for their research. It is only mentioned briefly in line 462.
Our Response:
Thank you for raising this topic. We have added a paragraph regarding these issues in the introduction. It is only needed to mention that on the one hand, there are not published materials to reference to about the cultural barrier and men dominancy cultures in Iran, and on the other hand, the objective of the paper was to identify the mechanisms of effects of mentoring on the women work-life balance and culture has been mentioned among the influential factors.
- Research Methodology
The study's sample consisted of Iranian women with at least 10 years of work experience who are managers, deputies, or specialists in a Tehran Municipality. This demonstrates that women hold top management positions in the aforementioned organisation.
- The demographic information (Table A1) did not indicate whether the participants were married or single. Referring back to my point in 1.b. above, culture may have an impact on Iranian women's employment participation. According to some studies, married Iranian women are less likely to enter the labour force, whereas highly educated women, married or not, tend to stay at their jobs for longer periods of time.
Our Response:
We understand the logic of your comment, and we understand that demographics can be an influencing factor. Therefore, a related paragraph is added to section 3.1 Sampling to provide a more complete picture of the demographic information of the participants in this study. However, we need to add that the purpose of this study is to observe the theory that emerged from the interviews, therefore we tried to have diversity among the participants to get a more complete picture of the theory that can be produced. That is why the participants were both married and single, men and women, mentors and mentees. For a cross-sectional study and when quantitative data is collected, only female mentees should be examined, and their marital status can be examined as an intervening variable. But in the study of grounded theory, the approach is different.
- The research should explain why these women in higher positions were chosen over those in lower positions. It would clear the air unless the current study intends to model a mentoring process specifically for women in top positions.
Our Response:
This comment is very relevant and correct, we also wanted to be able to cover all people at different levels of the organization who participate in mentoring programs. But since Tehran municipality is one of the pioneers of the mentoring program in organizations and since this program has been implemented recently, in the opinion of the organizational decision makers of the participants, only high-level positions were included in the mentoring program in the municipality. However, we have been told that this program is supposed to be expanded so that all municipal employees at all organizational levels can be included. Due to the importance of the issue and according to your comment, lines 272 to 276 have been added to the text of the article and the necessary explanations have been provided.
- Conclusions
Based on the current approach, the study did not highlight apparent limitations that was not being incorporated in their study which is the inclusion of culture and the exclusion of Iranian women in lower positions in developing the model.
Our Response:
This statement of yours regarding the impact of culture on women's employment and even on women's career path in Iranian society is correct, and according to your comment, we added a paragraph in the introduction section of the article in which we discussed this issue. However, the purpose of the paper is to develop a theoretical framework to identify the mechanisms of effects of mentoring on the women work-life balance and culture has been mentioned among the influential factors.
- Typography errors
Rectify your Table numbering.
Line 294 – should be Table 3
Line 303 – should be Table 4
Our Response:
Thank you very much. We noticed this mistake; the numbers and captions of the tables were corrected.

Reviewer 3 Report
Dear author(s). Thank you for your submission to Administrative Sciences. Your research idea has its charm. Unfortunately, the research design as well as the discussion and presentation of the research results do not meet the requirements of the journal.Please see some of gaps identified below:
Overall:
The structure of the paper is not clear and the content is not easy to comprehend - which might be as well due to the low English language level. The hypothesis are not clearly formulated.
Literature review / citations
The manuscript lacks heavily citations. The used literature is very limited and does show a critical analysis of existing theories.
The used definition is too general. Your research does not show which mentoring methodology (career or psychological orientation) the participating mentors use.
Methods
The methods parts offers a lot of definitions from the literature but shows serious gaps:
Why did the author(s) chose 3 mentors in relation to 17 mentees?
How many mentees relate to each of the mentors?
What is the hierarchical dependency between mentor and mentee?
As the group is so small the participating mentees might answer (maybe based on cultural factors) in a very positive way as anonymity is very limited.
Data analysis
The analysis does not show any control variables or control mechanisms to identify biases or potential risks of low anonymity. Your analysis does not show differences between mentors' responses and mentees' responses. What are the deviations between participants?
Logic of argumentation
The manuscript has some logic mistakes. One example is in line 159/160: "the change of circumstances due to globalisation" is not the major reason for organisations to focus on work-life balance. For example how did globalisation have an impact on work-life balance focus in the Tehran Municipality?
What are the cultural implications in this research how can the research results be generalised? Based on the presented research the application value is extremely low.
Ethical consideration
The manuscript lacks a statement about informed consent of the participants.
Author Response
Dear Respected Reviewer,
First of all, we would like to thank you for reading our work and for providing valuable comments on our work. We agree with all your comments, and we strongly hold the belief that your comments help us to improve the quality of our work. Following you can find our response to your comments:
Your Comment:
The structure of the paper is not clear and the content is not easy to comprehend - which might be as well due to the low English language level. The hypothesis are not clearly formulated.
Our Response:
Thank you very much for this comment. The current study is a grounded theory study, which is an exploratory study. The main goal of grounded theory is to discover a theory that lies in the layers of data. Therefore, in such studies, the authors should refrain from any prejudice, hypothesis, and theoretical framework so that they can identify and discover the relationships that appear in the collected qualitative data without any bias. Therefore, this study only has research questions presented in the introduction section, and the presentation of the hypothesis is not applicable in all grounded theory studies.
Literature review / citations
Your Comment:
The manuscript lacks heavily citations. The used literature is very limited and does show a critical analysis of existing theories.
Our Response:
We have tried to review as many related studies as possible. As you mentioned, the number of related studies in the literature is limited, especially the studies that have been conducted in Iran. Nevertheless, the second section of the article and table one, as well as the text after Table 1, are dedicated to the review and analysis of the literature on the subject. However, new references are added to the body of the text.
Your Comment:
The used definition is too general. Your research does not show which mentoring methodology (career or psychological orientation) the participating mentors use.
Our Response:
The question is very appropriate. This study is only about mentoring programs for the development of human resources in the advancement of the career path. Therefore, to answer your question, it should be said that the mentoring method is career orientation. Lines 67, 82, and 235 are among the parts of the article that mention this issue.
Methods
Your Comment:
The methods parts offers a lot of definitions from the literature but shows serious gaps:
Why did the author(s) chose 3 mentors in relation to 17 mentees?
How many mentees relate to each of the mentors?
What is the hierarchical dependency between mentor and mentee?
As the group is so small the participating mentees might answer (maybe based on cultural factors) in a very positive way as anonymity is very limited.
Our Response:
According to your comment, two paragraphs were added to the sampling section to provide a comprehensive picture of the participants in this study. We tried to answer your questions in these two paragraphs. It should be noted that the purpose of this study is to observe the theory that is obtained from the interviews, so we tried to have diversity among the participants to have a more complete picture of the theory that is produced. Thus, the participants were both married and single, men and women, mentors and mentees. For a cross-sectional study and when quantitative data is collected, only female mentees should be examined, and their marital status can be examined as an intervening variable. But in the study of grounded theory, the approach is different.
It should be also noted that, as explained in the text, we reached the saturation point in the interview 11, and the next nine interviews were conducted only to increase the reliability of the findings. The saturation point means that no new code appeared after the 11th interview and all the topics raised in the interviews were repeated and no new aspect of the phenomenon under study was presented.
Data analysis
Your Comment:
The analysis does not show any control variables or control mechanisms to identify biases or potential risks of low anonymity. Your analysis does not show differences between mentors' responses and mentees' responses. What are the deviations between participants?
Our Response:
As mentioned above, the purpose of the study is to find a theory from the interviews, and the purpose of the cross-sectional methodology is different from the grounded theory method. In the cross-sectional method, different variables can be considered as intervening or moderating variables and their effects on the dependent variable can be investigated. This is while the purpose of ground theory study is only to find different aspects of the phenomenon under study. It should be noted that the proposed theoretical framework of this study presented in Figure 1 should be quantitatively tested in subsequent studies. This is why the current study calls it a theoretical framework and we do not claim to present a confirmed conceptual model.
Logic of argumentation
Your Comment:
The manuscript has some logic mistakes. One example is in line 159/160: "the change of circumstances due to globalisation" is not the major reason for organisations to focus on work-life balance. For example how did globalisation have an impact on work-life balance focus in the Tehran Municipality?
Our Response:
Thank you very much, we tried to revise these sentences. You can see these amendments in the text. For example, the sentence you said was changed as follows:
However, in order to increase the performance (Beauregard and Henry 2009, Lazar et al. 2010) and satisfaction (Abendroth and Den Dulk 2011, Haar et al. 2014, Rani and Mariappan 2011) of the employees, organizations are continuously looking for solutions through which they can create a balance between the work and life of their employees.
((Abendroth and Den Dulk 2011) Abendroth, Anja-Kristin, and Laura Den Dulk. 2011. Support for the work-life balance in Europe: The impact of state, workplace and family support on work-life balance satisfaction. Work, employment and society 25: 234-56.
((Beauregard and Henry 2009) Beauregard, T Alexandra, and Lesley C Henry. 2009. Making the link between work-life balance practices and organizational performance. Human resource management review 19: 9-22.
((Haar et al. 2014) Haar, Jarrod M, Marcello Russo, Albert Suñe, and Ariane Ollier-Malaterre. 2014. Outcomes of work–life balance on job satisfaction, life satisfaction and mental health: A study across seven cultures. Journal of vocational behavior 85: 361-73.
((Lazar et al. 2010) Lazar, Ioan, Codruta Osoian, and Patricia Iulia Ratiu. 2010. The role of work-life balance practices in order to improve organizational performance.
((Rani and Mariappan 2011) Rani, Sakthi Vel, and Selvarani Mariappan. 2011. Work/life balance reflections on employee satisfaction. Serbian Journal of Management 6: 85-96.
Your Comment:
What are the cultural implications in this research how can the research results be generalized? Based on the presented research the application value is extremely low.
Our Response:
We agree with you on the impact of Iranian society’s culture on women's employment. Therefore, we dedicated a paragraph in the introduction of the article, in which we stated that the cultural issues of the society are a big obstacle in employment and even the career advancement of women. On the other hand, the findings of this research, which are presented in the form of a theoretical framework and depicted in Figure 1, also point to the importance of culture in creating a work-life balance for women.
At the same time, we also agree with you regarding the existence of limitations for generalizing the findings of this research. Therefore, we have discussed them in the conclusion section.
Ethical consideration
Your Comment:
The manuscript lacks a statement about informed consent of the participants.
Our Response:
The current study is a survey that studies mentoring processes and their impact on creating work-life balance. In other words, the subject of study is not human beings. Therefore, the Ethics Committee or Institutional Review Board is not applicable for this study. However, participation in this study was completely optional and has been done with the full consent of the participants (who are municipal employees). Besides this, the participants have been assured that all the collected information will remain anonymous, and this information will be used only for the purpose of conducting this study, i.e., studying the impact of mentoring processes on creating a work-life balance. And the collected data will not be shared with any person or entity.

Reviewer 4 Report
I would like to thank the authors for this research that aims seeks to model work-life balance of women in the mentoring process in Tehran Municipality.
The research subject is timely, innovative, and interesting. It also fits the aim and scope of the journal.
The research is well designed and follows a sound scientific research method.
In order to improve the quality of the research, several adjustments are needed.
Lines 64 and 64, you said : “. Tehran Municipality, due to its mission and organizational culture, needs a model to implement its training and development to achieve its goals”. Is women empowerment one of Tehran’s municipality goals?
Several sentences need to be reviewed/ shortened/ rewritten. Exp sentence line 71 and several others (see attachment).
Lines 76 to 91: Avoid bullets in this part of the research. Try to shorten the ideas and make then precise and concise. These are the research objective, so you can say: the first objective of the research (a) consists to provide a suitable model of …… Then you follow with objectives b, c and d.
Table 1/ add a meaningful title like the link (impact) between mentoring and work life balance. I also suggest that you make a reference to research that links between work life balance and remote work. This research could help you https://doi.org/10.1002/pa.2829
Lines 185- 191 : Avoid bullets and make a structured paragraph.
Research methodology: You need to mention when the interviews were conducted.
Line 283: To facilitate reading, you need to add a heading for this first paragraph.
Line 294: This is table 3, I guess? The title is wrong also as it talks about focused coding
Line 303: You need to add the title of table 4
The implications of your research are missing. You need to clearly indicate the contributions of your research?
It would be better to summarize your findings through a table or a diagram.
Other minor comments are directly attached to the manuscript.

Author Response
Dear Respected Reviewer,
First of all, we would like to thank you for reading our work and for providing valuable comments on our work. We agree with all your comments, and we strongly hold the belief that your comments help us to improve the quality of our work. Following you can find our response to your comments:
Your Comment:
Lines 64 and 64, you said : “. Tehran Municipality, due to its mission and organizational culture, needs a model to implement its training and development to achieve its goals”. Is women empowerment one of Tehran’s municipality goals?
Our Response:
Yes, indeed. Accordingly, we added related sentences into the introductions section.
Your Comment:
Several sentences need to be reviewed/ shortened/ rewritten. Exp sentence line 71 and several others (see attachment).
Our Response:
Thank you for your comments. We did accept all the comments you made in the attachment file. We tried our best to address all of them.
Your Comment:
Lines 76 to 91: Avoid bullets in this part of the research. Try to shorten the ideas and make then precise and concise. These are the research objective, so you can say: the first objective of the research (a) consists to provide a suitable model of …… Then you follow with objectives b, c and d.
Our Response:
Thank you for the comment, the text is revised accordingly.
Your Comment:
Table 1/ add a meaningful title like the link (impact) between mentoring and work life balance. I also suggest that you make a reference to research that links between work life balance and remote work. This research could help you https://doi.org/10.1002/pa.2829
Our Response:
The caption of the table is revised accordingly. Thank you for sharing the literature with us. We found it very interesting and related. Therefore, we added the findings of this paper in section 2.2 where we were talking about the literature on women work-life balance.
Your Comment:
Lines 185- 191 : Avoid bullets and make a structured paragraph.
Our Response:
It is revised accordingly.
Your Comment:
Research methodology: You need to mention when the interviews were conducted.
Our Response:
The interviews were conducted during a four-month course from mid-April 2022 to mid-July 2022. This sentence is added to main text in section 3.2 Data Collection Method
Your Comment:
Line 283: To facilitate reading, you need to add a heading for this first paragraph.
Our Response:
Thank you, this is the first paragraph of section 4. Findings.
Your Comment:
Line 294: This is table 3, I guess? The title is wrong also as it talks about focused coding.
Line 303: You need to add the title of table 4
Our Response:
We noticed that the order and the caption of the tables were written wrongly. We revised and corrected them accordingly.
Your Comment:
The implications of your research are missing. You need to clearly indicate the contributions of your research?
Our Response:
Thank you for the valuable comment. We, accordingly, added llines 630-645 in the conclusion section where we talked about the contributions of the study.
Your Comment:
It would be better to summarize your findings through a table or a diagram.
Our Response:
In fact, Figure one is the summary of the findings of this work.
Your Comment:
Other minor comments are directly attached to the manuscript.
Our Response:
Thank you for your comments. We did accept all the comments you made in the attachment file. We tried our best to address all of them.

Round 2
Reviewer 3 Report
Dear authors,
Thank you for your improved version of your manuscript.
I do not share your statement concerning informed consent and ethics approval. All primary research that include human beings (surveys and interviews fall in that category) need ethical approval and informed consent from participants.
As the research is extremely limited to Iran I advise to include this information in the title/subtitle.
Your grounded theory approach was used to identify a new theory. I lack to see this new theory and with that lack to understand the choice of your method.
Author Response
Dear Respected Reviewer,
Many thanks for your comments.
Here are your comments and our responses:
Your remarks:
Thank you for your improved version of your manuscript.
Our Response:
In fact, we appreciate your comments, and we do believe that they helped us to improve the quality of the work.
Your remarks:
I do not share your statement concerning informed consent and ethics approval. All primary research that include human beings (surveys and interviews fall in that category) need ethical approval and informed consent from participants.
As the research is extremely limited to Iran I advise to include this information in the title/subtitle.
Our Response:
Thank you for the comment. We have accordingly added a statement at the end of the article under the title of “Informed Consent Statement” (which is recommended by the journal of Administrative Sciences).
Your remarks:
Your grounded theory approach was used to identify a new theory. I lack to see this new theory and with that lack to understand the choice of your method.
Our Response:
The main objective of this article was to provide an understanding of how mentoring can be effective in creating work-life balance for women. Since there was no similar study in the literature, we did not have any hypothesis, and this was an exploratory study, which means there was no pre-determined idea to investigate different aspects of the subject under study. Therefore, we used the grounded theory method that allowed us to identify the mechanisms through which the mentoring program can affect women's work-life balance. The findings of this research, which are unique and emerged from the interviews, are presented in the form of a theoretical framework in Figure 1.

Reviewer 4 Report
The researcher made the suggested changes.
Author Response
Dear Reviewer,
Thank you for your kind consideration.
Authors